# Reproductive Ecology of the Chilean Round Ray (*Urotrygon chilensis*, Günther, 1872) in the Southern Gulf of California

Carlos J. Alvarez-Fuentes [1], Javier Tovar-Ávila [2], Jorge Payan-Alejo [3], Darío A. Chávez-Arrenquín [4], Isaias H. Salgado-Ugarte [5] and Felipe Amezcua [6,*]

1 Posgrado en Ciencias del Mar y Limnología, Universidad Nacional Autónoma de México, Joel Montes Camarena S/N, Mazatlán 82040, Mexico; cjalvarezfuentes@gmail.com

2 Centro Regional de Investigación Pesquera Bahía de Banderas, Instituto Nacional de Pesca y Acuacultura, Tortuga 1, La Cruz de Huanacaxtle, Bahía de Banderas 63732, Mexico; javier.tovar@inapesca.gob.mx

3 Facultad de Ciencias del Mar, Universidad Autónoma de Sinaloa, P. Claussen S/N, Mazatlán 82000, Mexico; payanalejo@gmail.com

4 Centro Regional de Investigación Pesquera Mazatlán, Instituto Nacional de Pesca y Acuacultura, Calzada Sábalo-Cerritos S/N, Mazatlán 82112, Mexico

5 Facultad de Estudios Superiores Zaragoza, Universidad Nacional Autónoma de México, Batalla 5 de Mayo S/N, Ciudad de México 09230, Mexico; isalgado@unam.mx

6 Instituto de Ciencias del Mar y Limnología, Universidad Nacional Autónoma de México, Av. Joel Montes Camarena S/N, Mazatlán 82040, Mexico

* Correspondence: famezcua@ola.icmyl.unam.mx; Tel.: +52-6699852845

**Abstract:** The Chilean round ray (*Urotrygon chilensis,* Günther, 1872) is commonly caught as bycatch in the Mexican Pacific, but changes in its reproductive ecology as a consequence of fishery effects have never been investigated. In this work, the reproductive ecology of this species was determined in the Southern Gulf of California (SGC). Total length (*TL*) ranged from 11.4 to 51.6 cm, and females were larger than males. Size at maturity ($TL_{50}$) was estimated at 27.5 cm for females and 25.3 cm for males. The seasonal variation of mature individuals, the presence of pregnant females, the mean oocyte size, the size of embryos and the smallest free-living specimens along the year suggest the existence of two reproductive periods during the year in the SGC, and the size of birth was estimated to be 11.2 to 15.6 cm *TL*. The average fecundity was 2.14 embryos. Spatial variations detected in size at maturity, fecundity, and reproductive cycle along the Mexican Pacific coast suggest the presence of separated populations. Some of these differences could be related to differences in maximum size attained in each region, though fishing pressure and environmental factors could have an important role, too.

**Keywords:** elasmobranchs; Urotrygonidae; reproductive rate; maturity curves; shrimp fishery bycatch; fecundity

**Key Contribution:** This work is the first to assess compressively the reproductive potential of the Chilean ray (*Urotrygon chilensis*) in the Mexican Pacific, where an intense trawl fishing activity takes place.

## 1. Introduction

The most prominent threat to elasmobranchs comes from commercial fishing [1], as they are caught indiscriminately in large quantities despite not being the primary target of fisheries in many cases [2]. This is the case for the Chilean round ray (*Urotrygon chilensis*) in the Mexican Pacific, which is an abundant benthic species with little economic importance [3,4] but which is commonly caught as bycatch in the shrimp trawl fishery operating in the area [5]. However, its abundance does not seem to be affected despite being commonly caught, probably because it shows a life strategy similar to organisms with an r strategy [6].

Possible spatial changes in the reproductive ecology of the Chilean Round ray as a consequence of fisheries effects have never been assessed. Nevertheless, high rates of fishing mortality might alter the life cycle and life-history traits, and therefore the reproductive characteristics of populations [7–9]. If the removal of specimens from a population occurs over a long time, the demographic changes can be substantial and carry important ecological implications such as diminished reproductive potential [10,11], changes in stock productivity [12–15], and long-term changes in life-history traits [15]. In populations where fishing mortality increases with size, early maturation may help ensure future egg production and population stability and potentially extend reproductive life [16]. However, this may compromise growth, making smaller specimens more vulnerable to predation [17].

Understanding the effects of fishing in exploited populations is important for both management and conservation, as intensive commercial fishing is a factor with the potential to critically affect fish-stock biomass and productivity, specifically if the exploited species is listed as Near Threatened under criteria A2d of the IUCN red list, implying that their population is decreasing. With this in consideration, our study sets out to examine the spatial variability in the reproductive ecology of the Chilean round ray in the Southern Gulf of California (SGC), which is the most important area in terms of fisheries in the country. The main goals are to determine the length structure of the population, to determine which parts of the population are being extracted by the trawl fishery operating in these areas, as well as the key reproductive parameters such as size at maturity, reproductive cycle, length at birth, gestation period, fecundity and sex proportion at birth in the area. The working hypotheses are that fishing operations affect specimens at different stages of maturity, in different fishing areas, depths, and seasons and that the species' reproductive strategy might have helped maintain stable population despite the high fishing mortality.

## 2. Materials and Methods

### 2.1. Studied Area and Sampling Procedure

The SGC is located within the Tropical Eastern Pacific Biogeographic Region (TEP) and corresponds to the two northern biogeographic provinces within this region. Specifically, the SGC (26°01′ N, −109°26′ W to 21°16′ N, −105°16′ W) comprises the southwestern part of the Cortez Province, corresponding to the GC up to Topolobampo, the 'Sinaloan Gap', that stretches from Topolobampo to Mazatlán, which is a primarily sandy/muddy coastline about 370 km in length interspersed with mangrove-lined lagoons, and the northern part of the Mexican province (Figure 1).

Data for this study originate from individuals of Chilean Round rays caught from research trawl surveys and small-scale fisheries undertaken in the area. The trawl surveys were undertaken onboard the R/V INAPESCA I and the R/V BIP XII belonging to the National Fisheries and Aquaculture Institute of Mexico (INAPESCA) for monthly intervals from September 2011 to September 2020. These vessels were fitted with commercial double trawl nets of 33 m and a mesh size of 2″. The trawls were undertaken in previously established positions at depths of 9–41 m, with a trawling duration of 1 h at a speed of 2 knots (Figure 1). From every fishing operation, a proportion of the specimens of the species obtained during the trawling were selected opportunistically once landed over the deck. All the specimens were frozen on board for their posterior analysis in the laboratory.

Additional samples from the small-scale fishery were obtained in three fishing localities within the region (La Reforma, Mazatlán, and Chametla) (Figure 1), in bimonthly periods from 2017 to 2020. This fishery employs gillnets of 700 m in length and 5″ of mesh size, at depths of 4 to 20 m. After every fishing operation, specimens were put on ice and frozen for posterior analysis.

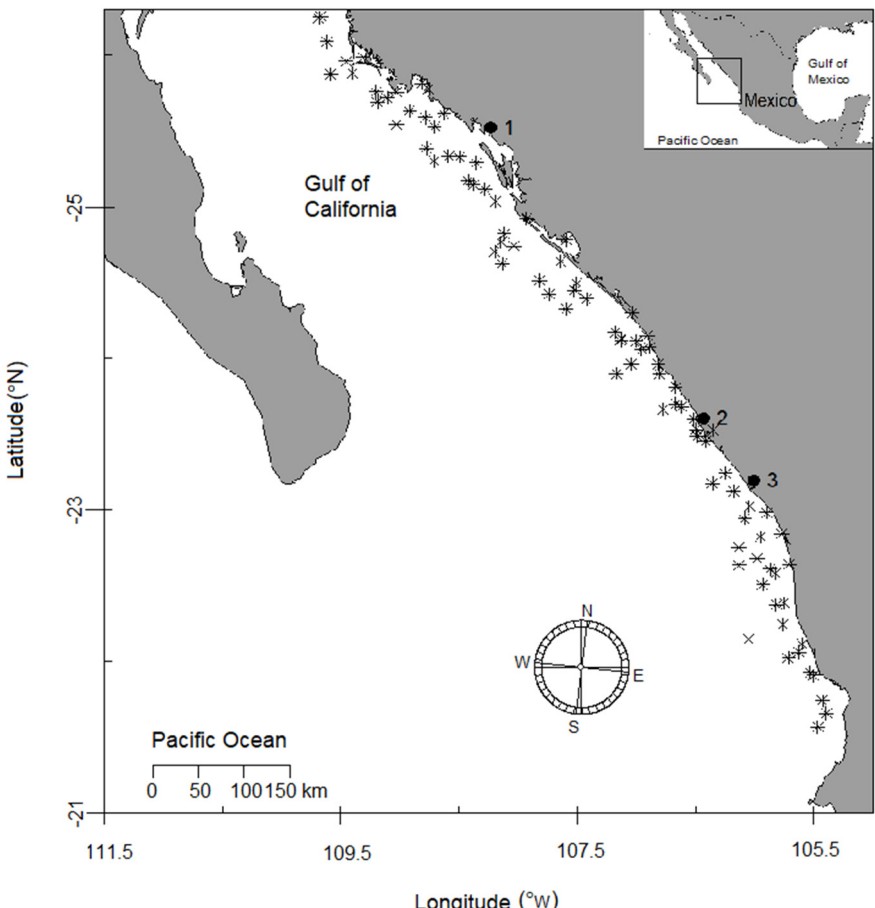

**Figure 1.** Area of the study showing the position of research trawls (asterisks), and the small-scale fishery grounds where samples were obtained (dots; 1. La Reforma, 2. Mazatlán, and 3. Chametla).

Analyzed specimens were dissected, sex identified, and total length (*TL*) was measured to the nearest cm. The mean *TL* for males and females and their standard deviation were calculated. The gutted weight and the weight of gonads and liver were obtained to the nearest 0.1 g. The proportion of females to males in the total sampling was estimated and compared to a proportion 1:1 with a Chi-squared test ($\chi^2$).

### 2.2. Length Analysis

A length–frequency histogram for males and females was made using the Sheater-Jones bandwidth estimator in R. A generalized linear model (GLM) was used to estimate the effect of sex, season, depth, and their interactions over the mean *TL* (cm) of the Chilean round ray. The seasons were divided as follows: dry cool season (DCS, from 1 December to 31 March), dry warm season (DWS, from 1 April to 30 June), and humid warm season (HWS, from 1 July to 30 November) [5]. The depth was divided into the following groups: <5, <9, <15, and >15 m. Data were checked for homoscedasticity using Cochran's C test. If significant differences were detected, a multiple comparison Tukey HSD test was used.

### 2.3. Reproduction
#### 2.3.1. Reproductive Cycle

The maturity stage of each individual was determined following the stages proposed by [7] but with modifications for this species, together with the criteria proposed by [8,18] which consider the following stages: GS = gonadic stage, OGS = oviducal gland stage, US = uterus stage, CS = clasper stage. In females, the number of embryos detected in the uteri and their sex was recorded. For the males, the length of the claspers (CL) was measured with calipers to the nearest mm from the pelvic fin axile to the tip of the clasper.

The reproductive cycle was determined through the seasonal variation in the maturity stages, the presence of gravid females in the catches along the year (monthly and seasonally), the size of the embryos and free-living specimens close to the size of birth, and monthly and seasonal variations in the gonadosomatic [$GSI$ = (gonad weight/gutted weight) $\times$ 100] and hepatosomatic [$HSI$ = (gonad weight/gutted weight) $\times$ 100] indices. Differences in the size of embryos along the year, $GSI$, and $HSI$ were determined with an analysis of variance (*ANOVA*) and a *Tukey HSD* post hoc test. The homogeneity of variances was tested with Cochran's *C* test.

### 2.3.2. Size at Maturity

The size at maturity ($TL_{50}$) was determined for each sex based on the proportion of immature and mature specimens at each interval class. Two sigmoidal models that were used to estimate $L_{50}$ were compared. The models were selected to represent two different maturity curves, asymmetric (Gompertz), and symmetric (Brouwer and Griffiths), and avoiding the use of redundant models [9].

Model 1 (Gompertz 1825):

$$P_i = e^{-e^{-\theta(TL_i - L_{50})}}.$$

Model 2 (Brouwer y Griffiths 2005):

$$P_i = \frac{1}{1 + e^{-(TL_i - L_{50})/a}},$$

where $P_i$ is the proportion of mature specimens in the length class $i$, $\theta$ is the rate at which sexual maturity is attained, $TL$ is the total length, $TL_{50}$ is the length at which 50% of the specimens are sexually mature and $\alpha$ is the width of the maturity ogive.

The models were fitted to the data using the minimum squares method and compared with the Akaike information criterion (*AIC*), the *AIC* differences ($\Delta_i$), and *AIC* weight ($w_i$) [9,19]. The difference between the ogives of both sexes was determined with an analysis of residual sums of squares [20,21].

### 2.3.3. Length at Birth, Gestation Period, Fecundity, and Embryos Sex Proportion

The length at birth was estimated as the average length of the largest embryos and smallest free-living specimens. The gestation period was estimated by the presence and size of recently fertilized oocytes and embryos throughout the year. Fecundity was estimated by counting the embryos found in the uterus. The proportion of embryos of each sex was compared to the proportion 1:1 with a $\chi^2$ test.

All statistical analyses and modelling were performed on RStudio 2022.12.0 + 353.

## 3. Results

A total of 606 *U. chilensis* specimens caught in the SGC during 2011–2020 from both small-scale and industrial shrimp trawl fisheries were analyzed. Though samples were not obtained during all the months of each year, an annual cycle was formed by adding the monthly samples of all the studied periods. Females ($n$ = 320) presented $TL$ from 11.4 to 51.6 cm ($\bar{x}$ 31.08, $s$ = 7.27), and males ($n$ = 286) a $TL$ from 11.2 to 41.3 cm ($\bar{x}$ = 30.46, s = 6.92). The proportion of females to males (1.12) was not significantly different from a 1:1 proportion ($\chi^2_{1,606}$ = 1.91, $p$ = 0.17).

### 3.1. Length Analysis

According to the GLM, the mean $TL$ of *U. chilensis* was determined by the three analyzed factors, sex ($F_{1,609}$ = 30.47, $p <= 0.01$), season ($F_{2,609}$ = 43.32, $p < 0.01$), and depth ($F_{3,609}$ = 26.05, $p < 0.01$). Females (mean $TL$ (cm) = 33.02 $\pm$ 7.47) were larger than males (Figure 2) ((mean $TL$ (cm) = 28.98 $\pm$ 6.51). According to the Tukey *HSD* test, specimens caught during the DCS (mean $TL$ (cm) = 34.16 $\pm$ 6.75) were larger than those caught during both warm seasons (DWS mean $TL$ (cm) = 30.99 $\pm$ 4.69 and HWS mean

*TL* (cm) = 29.79 ± 8.28, *p* < 0.05)). No differences were detected between the mean size of specimens caught during both warm seasons (*p* > 0.05). Organisms from the depth were larger than those from shallow areas; rays inhabiting depths deeper than 9 m were larger (<15 m mean *TL* (cm) = 33.9 ± 5.77; >15 m mean *TL* (cm) = 34.19 ± 5.87, *p* < 0.05), mostly mature organisms compared to those inhabiting depths less than 9 m (<5 m mean *TL* (cm) = 30.94 ± 5.96; <9 m mean *TL* (cm) = 29. 62 ± 4.02), which were immature organisms. No differences were detected in the mean *TL* between organisms inhabiting depths up to 9 m, and neither were detected between organisms inhabiting at depths from 9 m and deeper (*p* > 0.05).

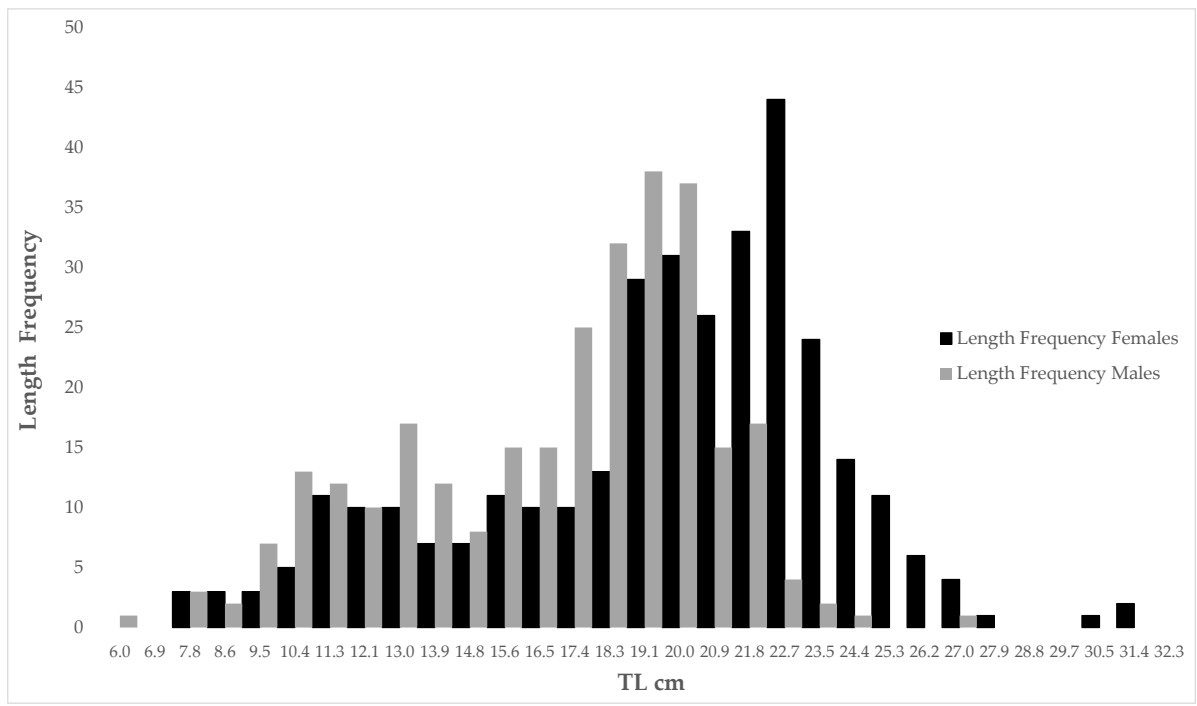

**Figure 2.** Length frequency of males and females *U. chilensis* in the SGC.

### 3.2. Reproduction

### 3.2.1. Reproductive Cycle

A large proportion of the sampled specimens was classified as mature for both males (71.7% of 286, GI = 3, and/or CI = 3) and females (75.3% of 320, GI = 3, UI ≥ 3 and/or OGI = 3) (Table 1). The largest number of mature specimens for both sexes was discovered during July and August. August was also the month when the majority of immature and mature specimens were located (Figure 3).

Mature males and females were discovered in higher numbers during the summer, but some peaks could also be seen in the winter, mainly in December for males and January–February for females (Figure 3). Gravid females (*n* = 107) were observed in the catches during most of the year, except for May, September, and October. The largest number of gravid females was also registered during July and August (*n* = 55, 41.8% of the monthly catch, and *n* = 46, 26% of the monthly catch, respectively), which corresponds to the humid–warm season (*n* = 66). Nevertheless, gravid females were observed during all seasons.

A total of 182 embryos were registered. The smallest embryos were observed in June, July, and August (5.1–5.5 cm of *TL*). In contrast, the largest embryos were observed in January and August (both females of 15.6 and 15.5 cm of *TL*, respectively). Medium-size embryos were also observed in January (7.1 cm of *TL*) (Figure 4). Significant differences were detected among the average embryo's monthly length (ANOVA $F_{5,185}$ = 7.80, *p* < 0.01). According to the post hoc Tukey test, significant differences (*p* < 0.05) were de-

tected between January with June, July, and August, which correspond to the early summer months. Two peaks in the maximum size of the embryos were observed throughout the year, the first in December–February (average = 12.5, 11.7, and 12.8 cm of *TL*, respectively) and a second one in June (average = 11.3 cm of *TL*) (Figure 4). The smallest free-living organisms were observed in February and April (16.5 and 13.5 cm of *TL*) and from July to September (16.5, 11.2 and 11.4 cm of *TL*).

**Table 1.** Maturity stages of *Urotrygon chilensis*, adapted from 7. GS = gonad stage, OGS = oviducal gland stage, US = uterus stage, CS = clasper stage.

| | Maturity Stage | Females | Males |
|---|---|---|---|
| 1. | Immature | (GS = 1) Ovaries are thin tissue strips with epigonal gland predominant and small white undeveloped follicles. <br> (OGS = 1) Oviducal glands are indistinct from the anterior oviducts. <br> (US = 1) Uteri are uniformly thin tubular structures. | (GS = 1) Testes are thin tissue strips with epigonal gland predominant. <br> (CS = 1) Claspers are small, and pliable with no calcification. |
| 2. | Maturing | (GS = 2) Ovaries are thickened strips with small white and yoking medium-size follicles. <br> (OGI = 2) Oviducal glands are distinct but only partly formed. <br> (US = 2) Uteri are thin tubular structures partly enlarged posteriorly. | (GS = 2) Testes are thickened strips with epigonal gland tissue extensive. <br> (CS = 2) Claspers are enlarged but partly calcified. |
| 3. | Mature | (GS = 3) Ovaries are enlarged and thickened containing large follicles with yellowish yolk. <br> (GOS = 3) Oviducal glands are heart shaped. <br> (US = 3) Uteri are uniformly enlarged and widened tubular structures but no eggs or embryos are visible (Sexually inactive). <br> (US = 4) Macroscopically visible eggs are present (sexually active). <br> (US = 5) Embryos of any size are present (sexually active). <br> (US = 6) No eggs or embryos but uteri walls are distended (in recovery or postpartum) (Sexually inactive). | (GS = 3) Testes are enlarged and predominant with epigonal gland tissue negligible. <br> (CS = 3) Claspers are enlarged, rigid and fully calcified. |

GSI and HSI

Though no significant differences were detected among the average monthly *GSI* of males (ANOVA $F_{8,135}$ = 0.93, $p$ = 0.48), probably due to the overlap of reproductive cycles of the species each year and low sample number during several months, two apparent periods of increment were observed along the year (Figure 5A). A peak was observed during March (average *GSI* = 0.69–0.78) and a second peak was observed during October–November when the highest value was reached (average *GSI* = 0.89–0.91). In May and December, the lowest male *GSI* was observed immediately after these maximums, respectively (average GSI = 0.48 and 0.6).

On the contrary, significant differences were detected among the monthly *GSI* of females (ANOVA $F_{7,153}$ = 23.80, $p < 0.001$). The mean *GSI* was significantly higher from March to May (average GSI = 2.17), and from November to January (average *GSI* = 2.27–2.86) (Figure 5A). The lowest *GSI* values were detected in February (average *GSI* = 0.96) and August (average *GSI* = 0.90), suggesting ovulation might have occurred during these months.

In terms of the *HSI*, no significant differences were detected in the mean monthly *HSI* value for males (ANOVA $F_{8,228}$ = 1.11, $p > 0.05$) (Figure 5B) or females (ANOVA $F_{9,263}$ = 2.31, $p > 0.05$) (Figure 5B). Both *GSI* of females and males, as well as the *HSI* of females, suggest the existence of two reproductive period parameters along the year in the SGC.

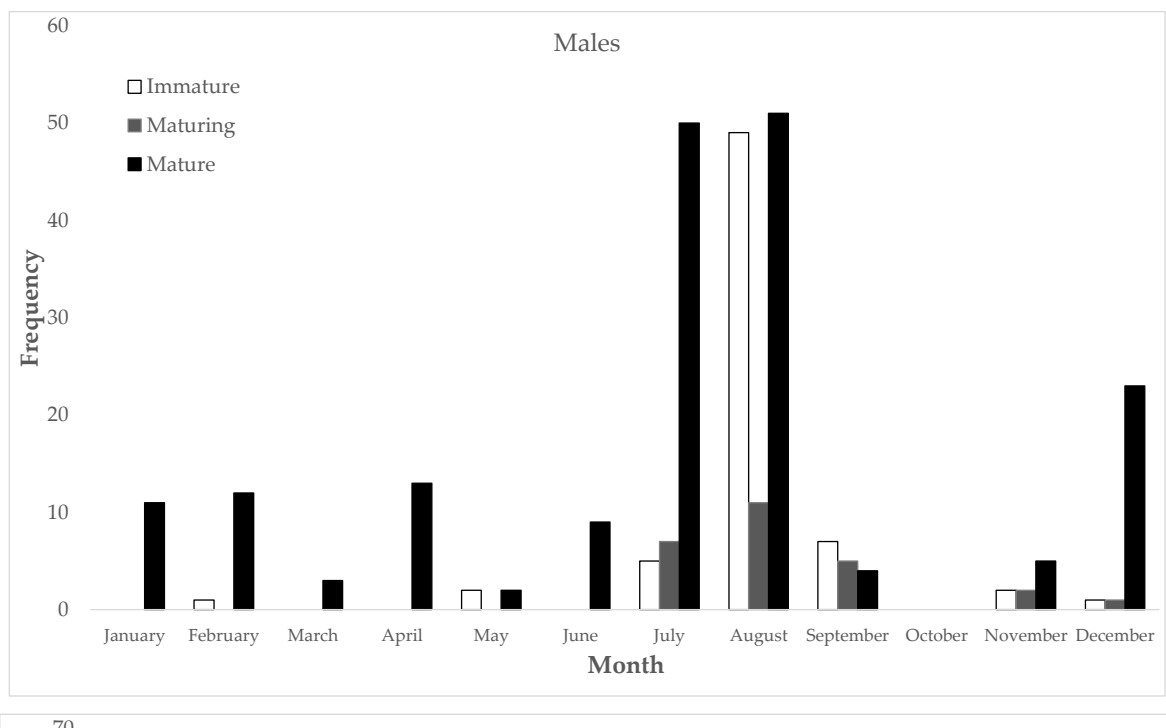

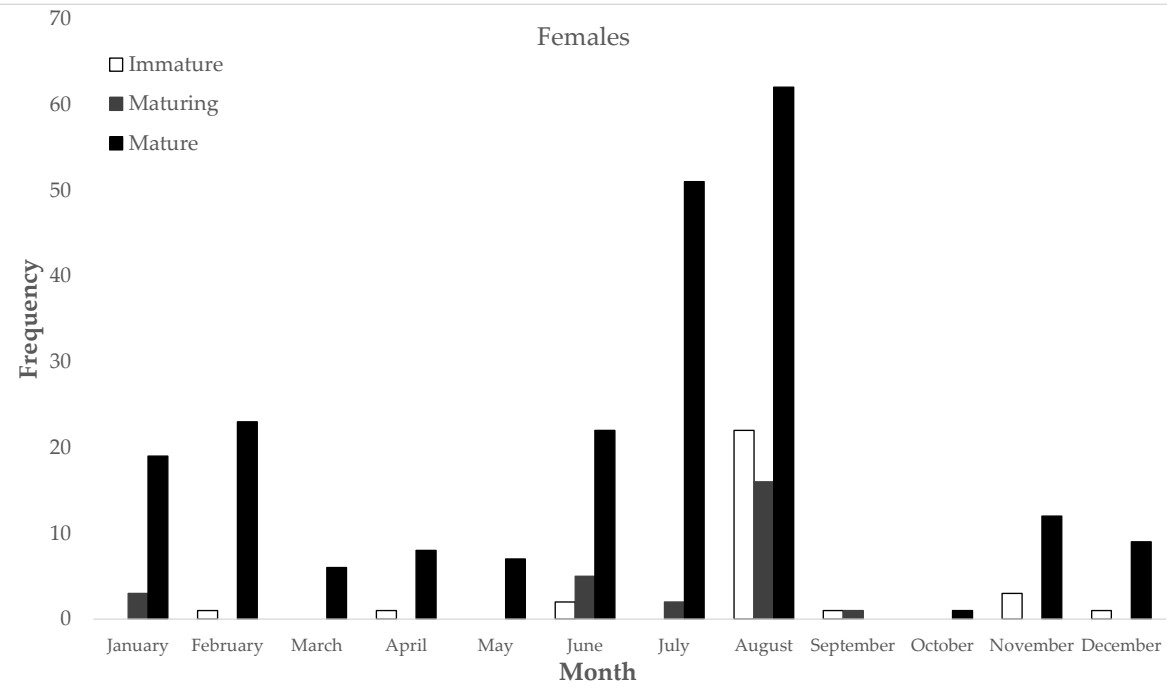

**Figure 3.** Seasonal variation in the maturity stages of *U. chilensis* in the SGC.

As occurred with *GSI* and *HSI*, the seasonal variation in maturity organisms, the presence of gravid females in the catches and embryo and free-living organisms size suggested a biannual reproductive cycle, with two breeding seasons followed by ovulation, the first one at the end of the DCS which corresponds to February–March, and the other one at the beginning of the HWS.

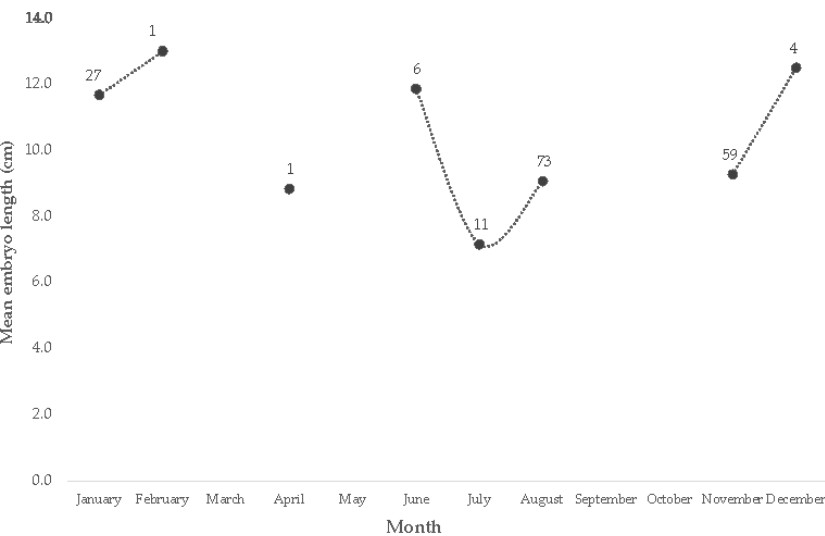

**Figure 4.** Average monthly size of embryos of *U. chilensis* observed in the SGC. The numbers represent the observations per month.

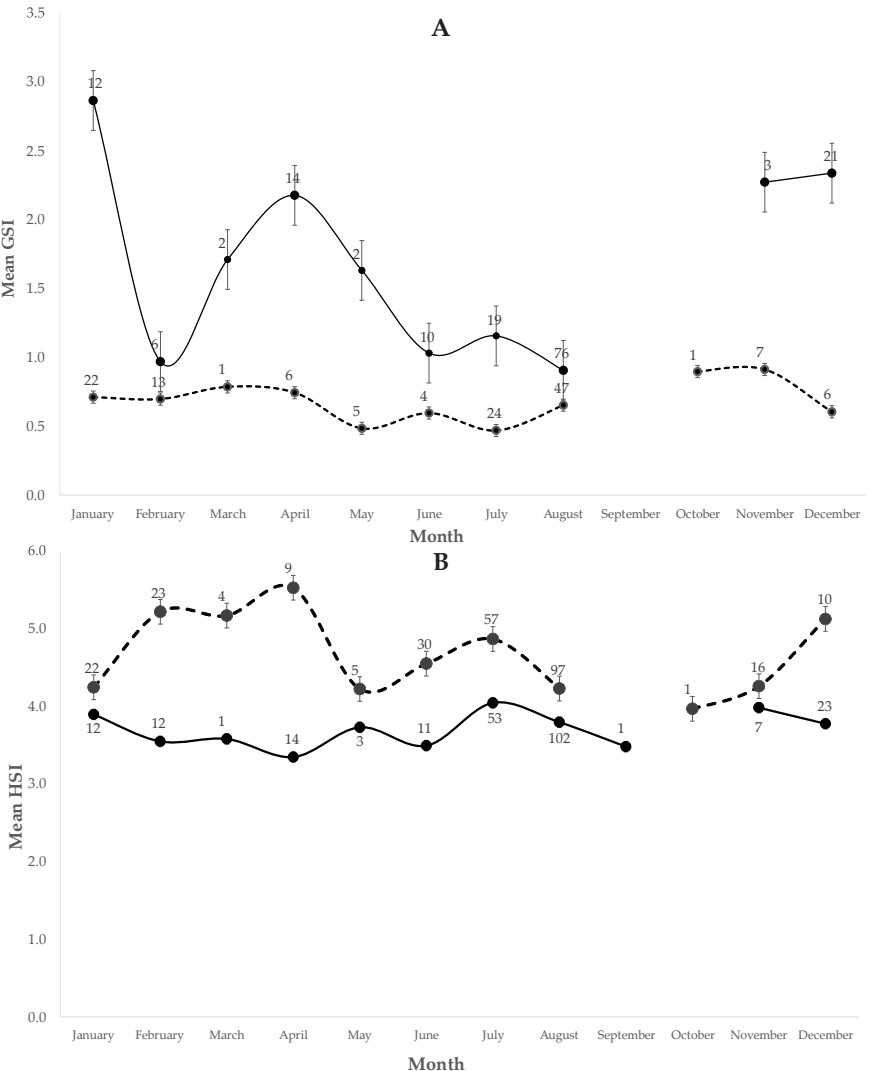

**Figure 5.** Gonadosomatic Index (**A**), GSI and Hepatosomatic Index (**B**), HSI for males (dashed line) and females (continuous line) of *U. chilensis* in the SGC. The bars represent the standard deviation, and the number is the N for that sex and month.

### 3.2.2. Size at Maturity

For males, the Gompertz model was selected as the best fit for the data according to Akaike. In contrast, the Brouwer and Griffiths model adjusted better to females (Table 2, Figure 6). Though the Gompertz model presented a considerable percentage weight of Akaike ($w_i\%$), its delta Akaike ($\Delta_i$) showed considerably less support according to the range proposed by [18] ($4 < \Delta_i < 7$). The size at maturity ($TL_{50}$) of females (Brouwer and Griffiths model) was slightly larger than that of the males (Gompertz model), though the $TL_{50}$ of the Gompertz model for females was lower. However, according to the Chen test, there was no difference between the Brouwer and Griffiths curves of both sexes ($F_{38,41} = 0.30$, $p = 0.8$), nor with the Gompertz curves ($F_{38,41} = 0.44$, $p = 0.7$).

**Table 2.** The size at maturity ($TL_{50}$) of *Urotrygon chilensis* in the Southern Gulf of California estimated with two models. The model selected as the best-fitted is marked with †. AICc = Akaike Information Criterion, $\Delta_i$ = Akaike's difference, $w_i\%$ = percentage weight of Akaike.

| Model | $TL_{50}$ (cm) | AICc | $\Delta_i$ | $w_i$ (%) |
|---|---|---|---|---|
| Females | | | | |
| Gompertz | 24.8 | −63.4 | 0.89 | 0.38 |
| Brouwer and Griffiths † | 27.5 | −64.3 | 0 | 0.61 |
| Males | | | | |
| Gompertz † | 25.3 | −65.1 | 0 | 0.96 |
| Brouwer and Griffiths | 26.3 | −58.5 | 6.63 | 0.03 |

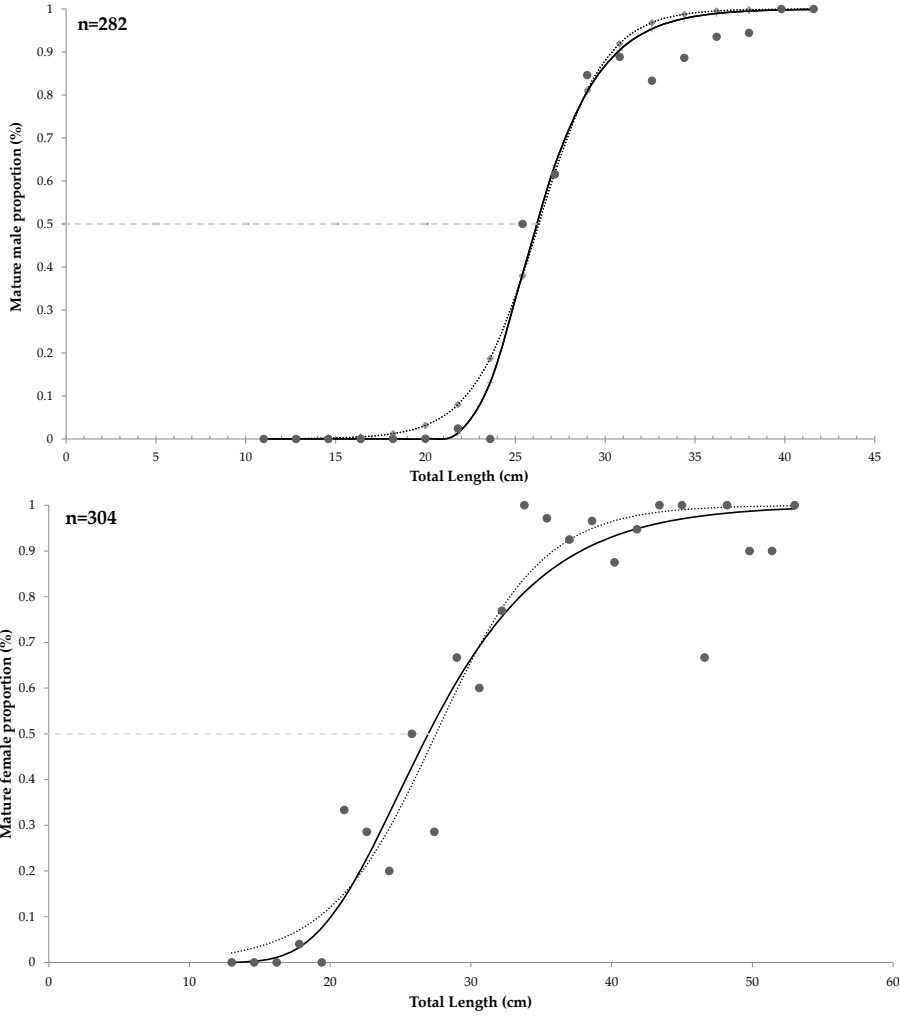

**Figure 6.** Maturity curves for males (**upper**) and females (**lower**) of *U. chilensis* in the SGC. The continuous line represents the maturity curve obtained with the Gompertz model, and the dashed line represents the maturity curve obtained with Brouwer and Griffiths model.

### 3.2.3. Length at Birth, Gestation Period, Fecundity, and Embryo Sex Proportion

The mean oocyte diameter was 15.1 mm, and varied from 4.3 mm in April to 24.2 mm during November, with the second largest observed during December (21.3 mm), although from June to August, large values were also observed. During the rest of the year, the average oocyte diameter was less than 14 mm (Table 3). Based on the largest embryo size (15.6 cm) and the smallest free-living individual (11.2 cm of *TL*), the size of birth was estimated to be 11.2 to 15.6 cm of *TL*. A gestation period of six months was estimated. In respect to fecundity, the number of pregnant females with one and two embryos were the most common, whereas the maximum number of embryos was six (observed in a female of 51.6 cm of *TL*). The mean fecundity was estimated to be 2.14 embryos (Standard deviation = 1.05). The sex of 232 embryos was registered, of which 49.6% were females and 50.4% males, thus no significant differences were detected in comparison to a 1:1 proportion ($\chi^2_{1,232} = 0.02$, $p = 0.09$).

**Table 3.** Monthly variation in the mean oocyte diameter of *U. chilensis* in the Southern Gulf of California. $\bar{x}$ = Mean, SD = Standard Deviation, *n* = Number of Individuals.

| Month | $\bar{x}$ (mm) | SD | *n* |
|---|---|---|---|
| January | 13 | - | 1 |
| February | 13.7 | 5.0 | 12 |
| March | 8.8 | 1.0 | 4 |
| April | 4.3 | 0.3 | 3 |
| May | - | - | 0 |
| June | 19.4 | 8.1 | 7 |
| July | 20.8 | 4.5 | 6 |
| August | 19.6 | 4.1 | 17 |
| September | - | - | 0 |
| October | 6.0 | - | 1 |
| November | 24.2 | 13.9 | 4 |
| December | 21.3 | 11.8 | 3 |

## 4. Discussion

Although the concern of the shrimp trawl fishery bycatch was raised since the beginning of the industrialization of the fishery, which happened at the end of the 19th century and its development during the 20th century [22], the recognition and evaluation of the vulnerability of elasmobranchs in this fishery are relatively recent [23]. Specifically, the capture of round rays in the Mexican Pacific has been reported since the late 1980s and early 1990s [24], with the inconvenience that landing information of these species is nonexistent because of their null commercial value, which means that they are usually discarded. Furthermore, information about the life history parameters of round rays and elasmobranchs in general in the Tropical Eastern Pacific Ocean is scarce, the consequence being that life traits that were uncommon for this group or abnormal conditions are only recently observed [25,26]. These factors preclude any attempt to understand the situation of these non-targeted exploited populations in order to establish management plans for such species.

Important parameters to consider when developing conservation and management plans for exploited populations are their reproductive parameters, because reproduction is the main contributor to stock restoration [27], and inevitably intensive commercial fishing is one factor with the potential to critically affect reproductive parameters [28]; thus, an understanding of these effects is important for fishery ecology in terms of both management and conservation.

Our results indicated that along the Pacific coast of Mexico, *U. chilensis* females are larger than males, as most other elasmobranchs [16,29] including other round rays of the family Urotrygonidae [17]. The fact that larger specimens were located during the cold season could be related to the oceanographic conditions observed during the winter; as the temperature decreases, probably together with the dissolved oxygen, larger batoids could concentrate in certain areas looking for optimal ecophysiological niches with warm waters rich in dissolved oxygen [30]. In terms of depth, larger organisms were discovered in shallow areas, and, in fact, these were caught by the small-scale fishery operating in coastal zones.

In terms of population variations, spatial size differences determined for *U. chilensis* from the SGC in comparison to other regions of the Mexican Pacific [31,32] (Table 4) could be related to population differences, although the depth interval at which captures were obtained could also have some effect. All these studies originate from the shrimp trawl fishery bycatch, and in similar depths: 13–43 m by [31], 13.2–55 m by [32], and 3–41 m in the present study, in which shallow samples from the small-scale fishery were also obtained. The smallest specimens in the Mexican Pacific were observed in the Gulf of Tehuantepec (GT), in the southern Mexican Pacific, and not in the shallower water samples at the SGC as mentioned before; on the contrary, the largest specimens were discovered in shallow areas in the SGC. Shallow areas were not sampled in the previously mentioned studies, but it seems that the shallow areas are preferred by the adult phases of this species.

**Table 4.** Comparison of the reproductive parameters of *U. chilensis* in different areas of the Mexican Pacific, including the present study (SGC). Ref. [30] (Michoacan and Guerrero, central Mexican Pacific, MG) and [31] (Gulf of Tehuantepec, southern Mexican Pacific, GT). Ref. [32] does not report $TL_{50}$ for the males nor average fecundity. SR = Sex Ratio; SM = Size at Maturity (cm); F = Fecundity; RP = Reproductive Period; BL = Birth Length (cm); SY = Study Year.

| Sex | *TL* cm | n | SR | SM | F | | RP | BL | Region | SY |
|---|---|---|---|---|---|---|---|---|---|---|
| | | | | | Avg. | Max. | | | | |
| ♀ | 14.5–44.5 | 100 | 0.96:1 | 25.2 | 1.7 | 4 | Spring | 14–15 | MG | 2004 |
| ♂ | 17.6–36.2 | 104 | | 26.5 | | | | | | |
| ♀ | 10.9–39.5 | 490 | 1:1/6.01:1 | 25.3 | | 5 | Spring/summer | 10–14.5 | GT | 2012 |
| ♂ | 10.9–33.8 | 211 | | - | | | | | | |
| ♀ | 11.4–51.6 | 320 | 1:1 | 27.5 | 2.14 | 6 | Winter/spring | 11.2–15.6 | SGC | 2022 |
| ♂ | 11.2–41.3 | 286 | | 25.3 | | | | | | |

Based on the results of this study, it can be concluded that in the SGC, this species has two reproductive seasons through the year: a major one during the summer, when the majority mature specimens were discovered and the oocytes were larger, and the other one during the winter, when there was also an important number of mature individuals and the oocyte diameter was the second largest. The existence of several reproductive periods along the year has been reported in rays of the families Urolophidae and Urotrygonidae, though most species appear to present annual or biannual cycles [33]. Nevertheless, a triannual cycle has been reported for *U. rogersi* [17], whereas a biennial and even a possible triennial cycle, with periods of resting and facultative diapause, has been proposed for *Urolophus cruciatus* [34] and an asynchronic cycle has been proposed for *U. viridis* [35], showing the diversity of strategies in these species.

It has been suggested that the length of the reproductive cycle could be related to the level of matrotrophy in these species [33]. For *U. paucimaculatus*, differences in the reproductive cycle between two regions in Australia have been reported, showing a biannual cycle in one region and an annual in another. Such differences have raised the question of the existence of two different species, but they instead have been associated with the effect of different environmental conditions and the large plasticity of the species [36,37].

In the same way, *U. chilensis* appear to present different reproductive cycles along the Mexican Pacific (Table 3), though it remains a question whether this could be related to environmental factors or fishing pressure.

The results of the *GSI, HSI*, and the gonadal development stage data can be used to reliably estimate the reproductive cycle [38–41]. However, in the present study, the possible concurrence of two reproductive periods prevented the observation of statistically significant female *GSI* and *HSI* peaks that could be associated with the reproductive seasons, though apparent graphic peaks were observed.

The mean size-at-maturity differed between males and females, not just in terms of size, but in terms of the model to which they were adjusted. The differences between sex could be related to the fact that males achieve maturity earlier, and do not reach the same sizes as females. For females, the $L_{50}$ is attained at approximately 53% of the maximum TL, whilst for males, $L_{50}$ is attained at 61% of the maximum reported length, so this difference is related to the sex-associated variations in size.

This estimated size-at-maturity was the largest for females in the Mexican Pacific (Table 4). Evidence of spatial variability in the size at maturity has been reported for other elasmobranchs, for example, between two populations of *Heterodontus portus-jacksoni* [40] and *Urolophus paucimaculatus* [36,37] in Australia. However, it has been stated that distortions of maturity ogives can be related to length selectivity by fishing gears, too [42]. Though the selectivity of trawl nets is assumed to be low, recent studies have shown that the use of some bycatch-excluding devices during shrimp trawls can produce selectivity effects on the elasmobranch species [43]. Furthermore, the criteria to determine the maturity stage of an organism based on macroscopic observation of reproductive organs could have an effect on the estimation of the size at maturity. In addition, it is necessary to consider that in the SGC, larger organisms obtained from the small-scale fishery were also analyzed; therefore, differences occurred in length selectivity as a consequence of different fishing gear, as previously stated, and, as such, a conclusion cannot be drawn.

The fecundity observed in the SGC was larger than the fecundity reported for MG. It has been stated that fecundity can increase as a result of fishing pressure, as a compensatory effect. A relationship between the size and fecundity has been reported previously in other round rays [17], and thus the fecundity detected in the SGC, where the largest females were caught, was as expected. However, this pattern was not followed by organisms caught in the GT [32] in comparison to the MG [31], a difference that could be related to the differences in the fishing effort among the regions.

The size of birth for *U. chilensis* is estimated to be 11.2 to 15.6 cm of *TL*, with a gestation period of six months. The size of birth was estimated from the size of the largest embryo and the size of the smallest neonate, which is a method previously used for similar species [44]. Considering that organisms of this size were captured, it can be said that the shrimp trawl fishery operates in nursery areas for this species. This hypothesis is supported also by the presence of gravid females bearing embryos near birth size in the area. Therefore, it is important to report these findings, because this fishery affects the population of this particular species in different ways [44].

Differences in size and some reproductive parameters, such as size at maturity and fecundity, can be detected between *U. chilensis* from the SGC and other regions of the Mexican Pacific, too (Table 4). These could also be related to environmental conditions, as has been determined in other elasmobranchs [16,40,44,45], including some rounded rays from the Western Pacific Ocean grouped in the family Urolophidae, separated nowadays from the American round rays, Urotrygonidae species [46], but with similar forms, sizes, and matrotrophic embryo development, such as *Urolophus paucimaculatus* [37] and *U. viridis* [34]. The differences detected in the reproductive parameters could also be related to fishing pressure. The SGC and GT are characterized by a high fishing pressure for elasmobranch species, particularly demersal rays, as both regions are important for the industrial trawl shrimp fishery in the Mexican Pacific [47]. On the contrary, trawling in the

CMP is rare due to the conditions of the sea floor in the region and the lack of abundant shrimp populations.

It has been stated as well that other factors could affect the estimation of such parameters, including differences in the depths and seasons of trawls, the author's criteria to define the reproductive parameters (e.g., the determination of maturity), or a combination of all the previous factors [42,48–50]. Previous studies have proved that increased fishing pressure in dioecious stocks could induce a reduction in overall population fecundity even in the absence of decreased fertilization rates. Such studies have indicated as well that high fecundity is correlated to rapid growth and vice versa [19,51,52]. If changes in the growth rate also occur between regions, that could explain the observed results, but estimating the growth was out of the scope of this paper.

The comparison of length and reproductive strategy of *U. chilensis* for different regions in the Mexican Pacific point out that the fishing effort is not affecting this species. In [53], the authors state that there has been an increase in the level of fishing effort on the populations of shrimp in the GT, resulting in a critical level of exploitation of some species, and this could have affected the *U. chilensis* population in the area. This shows a similar fecundity and $TL_{50}$ for the *U. chillensis* in GT. However, the higher fecundity in the SGC, as well as biannual reproductive cycle, could be related to the larger size of specimens in that region, the effects of fishing mortality, or environmental conditions. It is known that when a population is affected by fishing, as time passes, older (and larger) fish become fewer, because cohorts accumulate the effects of fishing mortality through time, so larger fish form a smaller proportion of the population [54]. In addition, high exploitation due to fishing leads to substantial modifications in the size structure of exploited communities, such as a reduction in size at maturity, and a decrease in the mean *TL* and mean maximum *TL* [54]. None of these effects could be detected in *U. chilensis* in the SGC since the largest specimens were observed in this region with a large fishing pressure. Differences in fecundity might be related to the size attained in the SGC instead of being an effect of fishing mortality. Nevertheless, the increase in reproductive events per year in the SGC and GT in comparison to the CMP could be related to an increment in fishing mortality.

It is necessary to be careful when drawing conclusions in respect to the population of this ray attaining sexual maturity at a smaller length, increasing the number of pups per or the number of reproductive events during the year as a direct result of fish harvest, as such consequences are difficult to determine for numerous reasons. Size selection patterns can vary over time due to the stochasticity of environmental conditions, and these can be also affected by variations in population contribution over time [55]. Size at maturity is influenced by many factors, including but not limited to the density of conspecifics, the density of other fish species, and ocean conditions [56]. Thus, the population effects of fishing cannot be revealed without careful consideration of the many factors affecting growth and maturation. The reproductive capacity of fishes is known to be strongly influenced by body length [57], and a population consisting of larger specimens, with consequently larger gonads, may thus be deemed to have greater reproductive potential than a population of smaller specimens. From the current results, it is apparent that in the SGC the population of *U. chilensis* had a higher reproductive capacity that might have help it to cope with the intense fishing mortality associated to the industrial shrimp fishery.

Assuming reproduction is the main contributor to stock restoration, the observed variability could significantly impact the conservation status for future generations of this population. In MG, where the effort is considerably lower than in the SGC and GT, it may be necessary to manage and preserve the reproductive stock of *U. chilensis,* whereas in those areas where elasmobranchs are commonly caught as bycatch, it would be necessary to evaluate the vulnerability of each species based on its susceptibility to be caught in order to develop effective management and protection measures [58]. In the SGC, it would be

necessary to consider the distribution of the reproductive stock analyzed in the present study according to the capture zone, seasons and depths.

While the risk of fishing in spawning grounds is higher during spawning periods, overfishing of immature specimens is potentially no less damaging, given that the sustainability and productivity of a fishery is dependent on the continued availability of juveniles. Management through the establishment of biomass closed season, such as the no fishing period of prawn fishery each year (May to September) and the elasmobranch fishery (June to July), may be sufficient for this species, considering that it seems to be secluded in the estuarine systems and targeted only by the gillnet fin fish fishery in a few regions. However, it is necessary to consider that small-scale fishers (such as those incidentally catching *U. chilensis* in the SGC) in tropical and subtropical zones of low institutional capacity countries are usually very poor and such measures may have significant economic and social implications. It is necessary to consider the economic and social aspects of the fishers as well as the ecology of target species before developing adequate and effective methods for fisheries management in the area.

## 5. Conclusions

From the current results it is apparent that in the SGC, the population of *U. chilensis* has a higher reproductive capacity that might have help it to cope with the intense fishing mortality associated to the industrial shrimp fishery. However, it may be necessary to manage and preserve the reproductive stock of *U. chilensis* in the area through the evaluation of its vulnerability and susceptibility to be caught, in order to develop effective management and protection measures [41]. For this, it is necessary to consider the distribution of the reproductive stock analyzed in the present study, according to the capture zone, seasons and depths.

**Author Contributions:** C.J.A.-F.: Conceptualization; Methodology; Data Curation; Formal analysis; Writing—original draft. J.T.-Á.: Conceptualization; Methodology; Data Curation; Formal analysis; Funding acquisition; Project administration; Writing—original draft; Editing. J.P.-A.: Methodology, Formal analysis; Writing—original draft. D.A.C.-A.: Methodology; Data Curation. I.H.S.-U.: Conceptualization; Methodology; Writing—review; Editing. F.A.: Conceptualization; Methodology; Formal analysis; Funding acquisition; Project administration; Writing—original draft; Editing. All authors have read and agreed to the published version of the manuscript.

**Funding:** The National Fisheries and Aquaculture Institute of Mexico (INAPESCA) funded the research surveys. The National Autonomous University of México (UNAM) through the Institute of Marine Sciences and Limnology (ICMyL), funded the laboratory research and analyses, and paid the processing charge fee of this article. The National Council of Science and Technology of Mexico (CONACYT) awarded the Ph.D. research grant 635764 to Carlos J. Álvarez-Fuentes.

**Institutional Review Board Statement:** All the samples were legally obtained from fishers with the appropriate fishing permits issued by the National Commission for Fisheries and Aquaculture.

**Data Availability Statement:** Data supporting reported results can be found at https://www.icmyl. unam.mx/uninmar/ (accessed on 1 March 2023).

**Acknowledgments:** We thank R. Cruz-Garcia and N. R. Leyva-Reyes for their help in processing the samples and sorting the database, and all the fishers that participated in the sampling program for their help in obtaining ray samples, and environmental data.

**Conflicts of Interest:** The authors declare no conflict of interest.

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
