# Peer review of "Reproductive Ecology of the Chilean Round Ray (Urotrygon chilensis, Günther, 1872) in the Southern Gulf of California"

_fishes, doi:10.3390/fishes8040193_

Round 1
Reviewer 1 Report
Dear Authors,
This paper present important information on the species U. chilensis about reproductive biology in southern Gulf of California, I have recommend that the manuscript undergo with all minor revision prior to its acceptance.
I mentioned deficiencies on the text.

Author Response
Ear Reviewer
Thank you for your comments. We have answered all your concerns, and actually, your review made us aware of some other parts that needed correction.
The species name was italicized throughout the text, TL50 was also corrected through the entire manuscript, and all the typos and inconsistencies were corrected. All the figures were redone, and the tables were revised.
Specific comments were answered in the attached pdf file.
We hope you find this new version ready to be accepted.

Reviewer 2 Report
I found this study well-focused on the reproductive life traits of an interesting fishing waste species. This is an important topic both for ecology and zoology, which deserve even more attention from the scientific community. Moreover, the manuscript was well-written and organized, and based on a good experimental design. Some key results were found and also limitations were well highlighted. Well done.
Please add in the title, and both the abstract and main text's first mentioning Urotrygon chilensis (Günther, 1872. Also, take care to italicize all the scientific names in the entire text, captions, etc.
Lines 152-154: this sentence is unclear, from the material and methods section I understood that sampling was carried out seasonally.
Lines 276-279: I strongly agree with this sentence, but to enhance the soundness of the manuscript it should be extended to a wide sense referring to elasmobranch in genera. Indeed, from the by-catch of shrimp fisheries, interesting data come out recently, regarding abnormal conditions or vary rare life traits, also linked to reproduction. See for example and to enrich this period:
10.3390/fishes7030120
10.1111/jfb.14468
Line 294: on which basis? There are different ecological theories in merit. Please argue.
Best regards
The Reviewer
Author Response
Dear Reviewer
We thank your comments that have much improved this manuscript. All of them were considered, and the text was changed accordingly.
Specifically:
We have added the authorship name to the title and the first mention of the species, and the species name was italicized throughout the text.
Lines 152-154: this sentence is unclear, from the material and methods section I understood that sampling was carried out seasonally.
Response: The M&M section was changed in order to agree with the results, in terms of the seasonality of the sampling program.
Lines 276-279: I strongly agree with this sentence, but to enhance the soundness of the manuscript it should be extended to a wide sense referring to elasmobranch in genera. Indeed, from the by-catch of shrimp fisheries, interesting data come out recently, regarding abnormal conditions or vary rare life traits, also linked to reproduction. See for example and to enrich this period:
10.3390/fishes7030120
10.1111/jfb.14468
Response:
Both papers have been reviewed, and the observation was included in the discussion. Also both references have been cited.
Line 294: on which basis? There are different ecological theories in merit. Please argue.
R: We have decided to delete this part, as we never establish this as an apriori hypothesis.